# Multiport Single Element Mimo Antenna Systems: A Review

**DOI:** 10.3390/s23020747

**Published:** 2023-01-09

**Authors:** Nathirulla Sheriff, Sharul Kamal Abdul Rahim, Hassan Tariq Chattha, Tan Kim Geok

**Affiliations:** 1Wireless Communication Center, School of Electrical Engineering, Faculty of Engineering, Universiti Teknologi Malaysia, Johor Bahru 81310, Malaysia; 2Advanced Cyclotron Systems Inc. (ACSI), Richmond, BC V6X 1X5, Canada; 3Faculty of Engineering and Technology, Multimedia University, Melaka 75450, Malaysia

**Keywords:** 5G communication, multi-element antenna, common radiator, single element, portable devices, SISO, wideband, MIMO, metamaterial, neutralization lines, parasitic material, mutual coupling, defected ground structure (DGS), planar inverted F antenna (PIFA), sub-6 GHz

## Abstract

In response to the increasing demand for voice, data, and multimedia applications, the next generation of wireless communication systems is projected to provide faster data rates and better service quality to customers. Techniques such as Multiple-Input–Multiple-Output (MIMO) and diversity are being studied and implemented to meet the needs of next-generation wireless communication systems. Embedding multiple antennas into the same antenna system is seen as a promising solution, which can improve both the system’s channel capacity and the communication link’s quality. However, for small handheld and portable devices, embedding many antennas into a single device in a small area and at the same time providing good isolation becomes a challenge. Hence, designing a shared antenna system with multiple feed ports with equivalent or better performance characteristics as compared to the approach of multiple antennas with multiple feed ports is a promising advantage which can reduce the size and cost of manufacturing. This paper intends to provide an in-depth review of different MIMO antenna designs with common radiators covering various antenna design aspects such as isolation techniques, gain, efficiency, envelope correlation coefficient, and size, etc. There is also a discussion of the mathematical concepts of MIMO and different isolation techniques, as well as a comparative analysis of different shared radiator antenna designs. The literature review shows that only very few antennas’ design with common radiator have been suggested in the available literature at present. Therefore, in this review paper, we have endeavored to study different antennas’ designs with common radiator. A comparison is provided of their performance improvement techniques in a holistic way so that it can lead to further develop the common radiator multiport antenna systems and realize the promising advantages they offer.

## 1. Introduction

With the explosive growth in users of present and future technology, there is a pressing need to design new compact antenna systems for higher data rate and large bandwidth. Design and implementation of Single-Input Single-Output (SISO) antennas for portable devices are easy and can be easily integrated into the portable systems. However, SISO systems are susceptible to the problems caused by multipath effect and for obtaining higher gain, the size of a SISO antenna should be increased proportionally [1]. For antenna to work efficiently at higher data rates in the existing frequency with an improved channel capacity is a great challenge. With the growing demand, designing Multiple-Input Multiple-Output (MIMO) antenna systems is a significant solution [2,3]. The MIMO systems precisely provide a technique in which multiple independent channels send and receive data concurrently in the same radio channel. The multiple transmit-and-receive antennas used in MIMO technology improve the radio link capacity to attain the multipath propagation. Next-generation wireless communication technology has presented MIMO systems as a feasible solution to solve the data rate limitation problem experienced by SISO systems, due to its multiple antennas’ capability which can also provide better system reliability and increased channel capacity [4,5]. The MIMO antenna system can enhance the signal quality and the gain of an antenna but leads to complexity in design and an increase in size [6]. To attain a required level of signal independence and good isolation in MIMO operation, the multiple antennas should be placed with adequate separation from each other. But for portable devices, this approach would need to be given more space for multiple antennas, as well as additional feedline length [7]. The close placement of the antenna elements for MIMO operations can solve the space issue but results in higher mutual coupling among antenna elements. 

In this perspective, various isolation techniques and different types of isolation enhancement techniques are found in literature to improve the isolation in various antenna structures which are discussed in detail in Section 2. Through these techniques, the gain, bandwidth, envelop correlation coefficient (ECC), and efficiency can be enhanced. The antenna diversity technique used in MIMO systems enhances the performance of MIMO systems through multipath fading and co-channel interference reduction [8,9]. Depending on the requirement, the diversity gain is achieved by employing different methods such as spatial, polarization, and pattern diversities.

Several findings regarding the potential antenna designs suitable for portable devices and small handheld terminals for the MIMO applications have been described in the literature. Existing literatures discussed in Section 4 have presented works on numerous MIMO antenna designs. In all these designs, more than one antenna elements are integrated or etched on single substrate to apprehend a compact MIMO antenna. This occupies more space and hence presents a big challenge in the industrial design process.

The shared radiator with multiport arrangement is a good alternative to miniaturizing the overall size of MIMO antennas. However, due to the difficulties in getting high isolation among connected ports, only a countable number of antenna designs having shared radiators with more than one port are available in literature. Compared to the array designs and separated ports, there is much less focus on common radiator designs with multiple feeding ports. To regulate mutual coupling, these coupled ports with the shared radiator require extra effort. Thus, designing an antenna becomes critical while considering the required parameters suitable for portable devices. 

Currently, we could not find any related review paper accessible in the literature which can provide us in detail about the shared radiator multiport MIMO antenna systems and their performance improvement techniques. So, in this paper, we have endeavoured to explore all the multiport MIMO antennas with a shared radiator, presented in recent years in a holistic manner considering their performance-enhancing techniques and aims to address the design issues for the further advancement in the MIMO antenna design as per their application. Our main aim in this paper is to provide a comprehensive literature review of multiport MIMO antenna systems having one shared radiator and a comparative study on different multiport-shared radiator antenna designs and the isolation improvement techniques used in these designs. The primary antenna features of shared radiator multiport MIMO antennas such as gain, frequency, mutual coupling, ECC and their applications are reviewed and compared in this paper. In addition, a brief overview of MIMO mathematical concepts and approaches used in these antenna designs for increasing the isolation are also discussed.

The rest of this paper is structured as follows. Section 2 discusses the mathematical concepts of MIMO, whereas a detailed literature review of diverse types of isolation techniques is elaborated in Section 3. The findings of single-element multiport MIMO antenna classification are summarized in Section 4. Section 5 then gives a detailed summary and discussion of the important findings of the paper.

## 2. Mathematical Concepts of MIMO

In this section, the mathematical concepts for MIMO antenna design are discussed [10]. Since the MIMO antenna system has multiple antennas at the transmit side (M_t_) and receive end (M_r_) as shown in Figure 1, the MIMO system capacity (C) can be expressed by Equation (1).
(1)$$C=E\left[\log_{2}\det\left(I_{\mathrm{Mr}}+\frac{P_{T}}{M_{t}+\sigma_{n}^{2}}\right)H\cdot H^{H}\right]$$
where P_T_—total input power, $\sigma_{n}^{2}$—noise power, H—channel matrix, H^H^—Hermitian transpose of channel matrix and I_Mr_—identity matrix.

Mutual coupling occurs when antenna elements are placed in closed proximity, which results in isolation reduction thus affecting performance. The envelope correlation coefficient (ECC) is required because the S-parameters such as S_12_ or S_21_ measured in between the ports are inadequate to encompass the effect of all S-parameters. The mutual coupling and return loss at the ports can be used to determine ECC, which helps to find the diversity performance of the MIMO antennas [11]. The acceptable and standard value of ECC should be less than 0.5 for portable devices. However, the ECC calculation through S-parameters is effective only in case of lossless antenna substrates but in case of lossy antenna substrate, the ECC must be calculated/measured from the antennas’ far-field radiation patterns. In Table 1, the acronyms and their meanings are included.

The mean effective gain (MEG) is an important performance parameter and can be defined as the ratio of the mean power received to the mean incident power of the antenna. It is used to calculate the average received signal strength of each antenna [12]. Another performance parameter, which is the total active reflection coefficient (TARC) is the ratio of square root of total reflected powers to the total incident powers [13,14]. The TARC is a function of frequency and is expressed in Equations (2) and (3).
(2)$$\Gamma =\sqrt{\frac{\mathrm{avialable}\;\mathrm{power}-\mathrm{radiated}\;\mathrm{power}}{\mathrm{available}\;\mathrm{power}}}$$
(3)$$\Gamma_{a}^{t}=\frac{\sqrt{\sum_{n=1}^{N}{\left|b_{n}\right|}^{2}}}{\sqrt{\sum_{n=1}^{N}{\left|a_{n}\right|}^{2}}}$$
where, [b] = [S][a] and [S] is the antenna’s scattering matrix, vector [a] is the sum of the available incident power at all the ports, vector [b] is the power reflected to the source and a_n_, b_n_ are the nth element of the vector [a] and vector [b] respectively.

Similarly, the antenna Diversity Gain (DG) is a well-known performance parameter used to verify the efficacy of the diversity. It can be defined as the ratio of rise in SNR of mixed signal from a multiple antenna to the SNR from a single antenna in the system. The DG can be calculated using Equation (4).
(4)$$\mathrm{DG}=\sqrt[{10}]{1-{\left|\mathrm{ECC}\right|}^{2}}$$

Like the above parameters, the dielectric substrate permittivity, loss tangent and thickness play an important role in designing the antennas. The efficiency, return loss and gain can also be improved by the proper selection of the dielectric material.

## 3. Isolation Techniques

Mutual coupling between the antenna elements is the energy absorbed by the antenna element in MIMO system when the nearby antenna is transmitting/receiving. As the energy radiated away from one antenna gets absorbed by a closely placed antenna, it becomes adverse. This affects the efficiency and performance of the antennas in both transmitter and receiver side.

Good isolation between the two antennas would be achieved by placing the antennas in such a way that they have sufficient space between them [15]. However, this required space between the antenna elements will increase the antenna size. Other than space, there are various isolation techniques which have been reported in literature for increasing the isolation among antenna elements.

### 3.1. Defected Ground Structure

Defected ground (DG) structure is an isolation technique in which the slots or slits are inserted on the antennas ground plane to change its electromagnetic properties. By modifying the ground plane [16] as shown in Figure 2, the isolation is improved. The ground plane modification can be in the form of slits [17,18] or tumble-shaped defects [19,20].

DG structure acts as a band stop filter by disturbing the current flow on the ground plane. The coupled fields generated between the nearby antennas are suppressed. The mutual coupling is reduced between the two antenna-radiating elements by periodic S-shaped-defected ground structure [21]. This disrupts the electromagnetic far-field drastically and the current is induced between patch elements. A square ring-defected structure discussed in [22] is added to the ground plane to act as a resonator. It decreases the cross-polar levels, and the surface waves are decreased by limiting them within the dielectric. The reduction in surface wave further helps to decline in back radiation which results in getting good isolation.

### 3.2. Decoupling Network

In [23], a decoupling network (DN) also termed decoupling and matching network (DMN) is used. It is a circuit-based approach to achieve good isolation in MIMO antennas. The basic principle of DMN is to reduce the coupling coefficients between the antennas while maintaining a good impedance matching for each antenna [24]. Figure 3 shows two L-shaped Inverted-F antennas which act as radiating elements. The diamond-shaped pattern etched in the ground known as the diamond-shaped pattern ground resonator (DSPRG), acts as a resonator which also works as a decoupling element [25]. This can control the surface current flow in the antenna elements and provides good isolation in the wideband.

In [26], another decoupling method known as a coupled resonator decoupling network (CDRN) is implemented between the two nearby antennas to achieve good isolation. CDRN consists of two similar open-loop square-ring resonators.

### 3.3. Electromagnetic Bandgap (EBG) Structure

An electromagnetic bandgap (EBG) structure is designed on the ground plane by periodic arrangement of unit cells. This structure acts as a band stop filter and provides high attenuation, which helps in achieving reduction in mutual coupling between the antennas [27,28]. Each cell in an electromagnetic bandgap structure looks like a mushroom inserted between two antenna elements. It can act as a band stop filter at different frequencies by adjusting the unit cell dimensions and its arrangement. 

### 3.4. Neutralization Lines

This is a decoupling technique in which a conductive wire or strip is used that links the two antenna radiating elements. The idea of a neutralization line is to minimize the coupling between the two antenna elements by establishing a new coupling path and improves the bandwidth when connected between ground planes [4].

Neutralization lines with various lengths can be utilized for achieving multi-bands or a wide band operation and to obtain high isolation. Good isolation is achieved for frequency from 1.7 GHz to 2.7 GHz using multiple neutralization lines as shown in Figure 4. Three neutralization lines are used with different lengths to achieve multi-band frequency coverage [29].

A neutralization line introduced between two antenna elements creates some current on the line path and generates an extra EM field to cancel the mutual coupling as discussed in [30]. Between the C-shaped radiator antennas, a neutralization line is incorporated in parallel. This line cancels out unnecessary coupling by producing opposite coupling and generates imaginary admittance [31].

### 3.5. Parasitic Elements

This decoupling technique can be implemented to reduce the mutual coupling by placing the parasitic elements in between the antennas in MIMO configuration by cancelling a segment of the coupled fields through an opposite coupling field, which can reduce the total coupling on the selected antenna. These elements are not coupled to the antennas in any way. The parasitic element can be of any type like a resonator or shorted stubs [32]. Parasitic elements are also used to adjust the bandwidth, isolation range, and coupling quantity [33]. A parasitic decoupling structure of a T-shaped stub with a slot is employed on the backside of the substrate between the two square monopole elements [34]. The antenna matching is improved through the stub and the slot attached to stub reflects the radiation from the elements. Hence, the isolation is improved. Two orthogonal modes [35] used in the parasitic elements creates a impedance bandwidth broader by coupling either in radiating patch or in the ground plane. Mutual coupling is minimized, and a good isolation is achieved by establishing new additional coupling path [36,37]. Additionally, good isolation is achieved by the extra coupling path resisting the signal approaching from the other path.

### 3.6. Complementary Split Ring Resonators (CSRR)

This method is one of the good solutions for mutual coupling reduction in which magnetic and electric coupling are produced from an LC resonator. This method is useful in filtering and improving isolation [38]. The impedance bandwidth is broadened through the proper placement of CSRRs on the patch at a symmetrical position of the MIMO antenna in which the two printed dipole antennas are perpendicular to each other [39]. Since the CSRRs behaves as negative permittivity band-stop filters, higher-order modes are suppressed. In [40], a slotted CSRR etched in the ground plane between the antennas achieves a good isolation. In this configuration, the length of slotted CSRR etched in ground plane is adjusted to improve the isolation. This technique is also known as space diversity utilization.

### 3.7. Metamaterials

Metamaterials are useful for the reduction of mutual coupling implemented between adjacent antenna elements and the presence of a band gap in their frequency response [33]. The decoupling component is made of sub-wavelength metal-air layers, which can be treated as a singular medium operating over a broad frequency band [41]. Four element substrate integrated cavity-backed slot antennas distributed orthogonally comprise two-layer mushroom structure and the isolation are improved by placing a wall between each antenna element. MIMO antenna with reversal composite right-left-handed configuration is discussed in [42]. Here, the slit and interdigital capacitors are jointly inverted by cascading the metamaterial transmission line. Hence, by this configuration, the current produced inside the antenna is reversed. This configuration reduces the mutual coupling, and a good isolation is achieved.

The minimum value of mutual coupling achieved in the above-discussed techniques is between −15 dB to −20 dB. The values lower than this can affect the antenna mismatches and embedded radiation efficiencies of the MIMO antenna systems.

## 4. Multiport Single Element System 

MIMO technology offers many benefits, including error-free and faster communication, as well as superior multipath propagation results. MIMO technology, which employs multiple trans-receive systems, also improves transmission and reception quality. The performance metrics required to characterize the behavior of multiport antenna systems is discussed in [34]. The author suggests that existing designs cannot fulfill the needs of future wireless handsets that can cover the 4G and 5G (upper and lower) frequency bands along with WLAN and WiMAX, while considering the wide bandwidth. Hence, novel designs should be implemented to satisfy the requirement.

For 5G MIMO antenna base stations, the 3-D printing technology and metal plating technique are used together to fabricate metal-coated polymer antenna [43]. The fabricated antenna has an isolation of less than −14 dB with antenna dimensions of 27.2 × 27.2 × 17 mm^3^. Four identical box-folded PIFA MIMO antenna, designed for mobile handsets with a wide operating band of 1.84–2.69 GHz. However, the whole antenna system dimensions are large and not the most suitable for portable devices [43].

A dual-band MIMO antenna for LTE Terminal is presented in [44]. To reduce the mutual coupling, the coupling feed patch and capacitive load patch are used in this system, which reduces the current density on the ground plane and the slotted technology is used to achieve the dual band. In contrast, a two-element four-shaped MIMO printed antenna system designed with defected ground structure and micro-strip matching network is used to enhance the isolation [45]. An eight-port antenna array with each pair of antennas is placed symmetrically on the substrate at the four edges. Each antenna element is autonomously coupled with a feeding strip and a communal square loop is etched at each antenna pair. Good isolation is achieved through this configuration by orthogonal polarization, but the efficiency is compromised because of the non-availability of the external decoupling structure [46,47].

A multiband antenna system for LTE application, discussed in [48], have 3D Inverted-F Antennas folded on a non-metalized section of PCB. A new resonance is created by inserting the parasitic radiating element to IFAs and high port-to-port isolation is achieved via a suspended neutralization line between the two ports. A novel MIMO antenna system with a nature-inspired flower shape is proposed for upcoming 5G communication systems [49], which covers the mmWave frequency band. In [50], a rectangular dielectric resonator antenna (RDRA) with circularly polarized response is designed which covers the frequency band from 4.4–4.8 GHz suitable for 5G NR (New Radio) Sub-6 GHz band applications. Similarly, an eight-element MIMO array structure is presented in [51], which covers the frequency range between 3.4 GHz to 3.6 GHz. This helps to surge the data throughput with intra-band contiguous carrier aggregation. This is a circular-monopole antenna for portable devices with each antenna element consisting of three microstrip transition printed on the top side of substrate and two L-shaped stubs at the ground plane to improve the isolation [52].

An eight-port planar antenna for 5G micro wireless access point applications is designed by integrating three monopole strips with different structures and resonant modes for a wider bandwidth. High isolation and low ECC are obtained through diverse protruded stubs and etched slots loaded on the defected ground plane [53]. In [54], a printed antenna is designed which is capable of multiband operation with coupled-fed monopoles providing good isolation of less than 15 dB over all the frequency bands. A hybrid antenna with 4G and 5G antenna modules combined operating at 3.5 GHz band capable of covering the C-band is designed [55]. Four-port MIMO antenna for sub-6 GHz 5G applications is designed on an FR4 substrate and the overall volume of the circuit board is 50 × 50 × 1.6 mm^3^ [56]. A compact dual-band 2-port antenna operating at 2.4–2.5 GHz works simultaneously with a 4-port antenna operating at 4.9–5.725 GHz, without using the MEMS switching. Isolation is better than 12 dB achieved through back-to-back slot arrangement [57].

The MIMO antennas discussed in [43,44,45,46,47,48,49,50,51,52,53,54,55,56,57] provides good isolation but their structure is either complex or their total size is larger. Hence, designing a compact MIMO system considering all the performance parameters is a challenging task for portable devices and mobile terminals. The approach of designing common radiator MIMO antenna with multiple feed ports providing better performance could represent a considerable advantage by reducing the size and the manufacturing cost. In the existing literature, it is observed that few studies have been conducted on MIMO multiport single element antenna systems. The following sections will present the recent literature regarding two-port or four-port antenna systems sharing a single radiating element. 

### 4.1. Single Element Two-Ports MIMO Antenna System

In [58], a novel single element MIMO antenna with dual-port based on isolated mode antenna technology (iMAT) was discussed. The phase and magnitude of the signal are changed due to few bends available in the crossover coupling structure connected to the single antenna radiating patch with two ports. To solve this issue and to attain good isolation, an I-shaped slot is etched at the ground plane which behaves as a LC filter. The iMAT concept was introduced in [59] also to achieve enhanced port-to-port isolation in which a U-shaped antenna with common radiator is designed and compared with two elements monopole antenna.

In [60], a common radiator dual port antenna using iMAT for LTE band application was designed. A good isolation was achieved by etching a rectangular slot at the ground plane on the bottom side; which intersects with a T-shaped slot inserted at the radiating patch as shown in the Figure 5. The multiple slots and slits etched on the top patch and the ground layer reduces the mutual coupling and has a good bandwidth covering the LTE 2300; 2.4 GHz WLAN and LTE 2500 frequency bands. This proposed common radiator dual port antenna has a dimension of 48 × 33.8 mm^2^ suitable for mobile devices. The single element MIMO antenna system presented in [61], consists of two wideband monopole antennas which are orthogonally placed with a shared radiator and L-corner ground plane for the frequency bandwidth of 600 MHz from 2.3 to 2.9 GHz. As shown in Figure 6a; a parasitic element of chain-shaped structure with two sets of rings are etched on the top and bottom layer of the substrate. The position of rings is adjusted in such a way to generate a secondary current path; which can reduce the current distribution effects between the ports. This configuration provides good isolation between the ports and a return loss of less than −10 dB is achieved.

The two-port MIMO antenna system in [61] is presented for wireless access point indoor applications. In this design, two coplanar waveguide-fed ports which are perpendicular to each other are connected to a common radiator. On the same side of the substrate, an L-shaped ground plane is also etched as shown in Figure 6b to excite two orthogonally polarized modes [62]. Good isolation is obtained by introducing the chain-shape parasitic element (CSPE) between the ports. The total dimensions of the antenna structure are 110 × 130 × 1.6 mm^3^. The stepped-cut technique at four corners is used by adjusting the shape of radiating patch to obtain the required bandwidth. The implemented antenna achieves a good isolation of around 15 dB, providing impedance bandwidth of 1.8 GHz from 0.9 GHz to 2.7 GHz [62].

A dual-port monopole antenna (64 × 64 mm^2^) attached to a common radiator with a shape of circular disc with a radius of 20 mm was implemented in [63]. To excite from each port, a feed line 20 mm long and 3.5 mm wide is connected. To expand 10 dB input impedance bandwidth, a groove is inserted below each feeding line of the antenna as shown in Figure 7. To achieve a good isolation, a circular cut of radius R_c_ was etched in the ground plane, which restricts the current flowing between the two ports. Wide bandwidth of 2 GHz to 6 GHz is achieved with less than −15 dB port-to-port isolation. Similarly, the dual-port microstrip antenna was presented in [64] which utilizes an external loop to attain good port-to-port isolation. An ACS-fed multiple-input–multiple output antenna presented in [65] operates from a frequency range of 3.1 to 10.6 GHz. A good isolation of more than 15 dB is achieved. The antenna design discussed in [66] is much smaller in size of 26 × 26 mm^2^ as compared to the antenna structure presented in [65]. This size reduction was achieved by two antenna elements sharing the single radiator and the ground size is reduced using an ACS-fed structure as shown in Figure 8a. The operating bandwidth and the isolation between the two ports are enhanced by attaching a rectangular patch on the bottom side and etching a I-shaped slot in the radiator.

A modified common rectangular radiator connected with two antenna elements is fed by two microstrips placed perpendicular to each other and the ground plane connected by a short strip as shown in Figure 8b [66]. The current flow between the two antenna elements is reduced and the isolation is improved through the insertion of slots in the radiator. This configuration disturbs the coupling by focusing the current flow on the edges of the slots. Good isolation of less than -15 dB is achieved with a reflection coefficient of better than 10 dB from 3 GHz to 11 GHz.

The MIMO antenna design shown in Figure 9b as an ultra-compact size of 22 × 24.3 mm^2^, which is smaller as compared to the designs presented in [65,66,67]. Figure The design discussed in [68], the single shared radiating element is fed with two perpendicular meandered microstrip lines. Good isolation is obtained for the entire bandwidth through the placement of an open shunt stub on the radiator and a T-shaped slot in the middle of the radiator. The use of meandered slots reduces the total length while maintaining the electrical length of the microstrip lines which helps to achieve compact size of the antenna [69] but increases the energy leakage. Hence, a partial ground plane of small strips is placed on the bottom of the substrate and the stub connected to the ground provides the current a path to the stub providing good isolation between the two ports. The antenna presented in [70] consists of a common radiating element with leaf-shaped structure, shared evenly as shown in Figure 10, using two microstrip lines and a common ground plane with curved shaped slot etched on the bottom side of the substrate. Isolation less than −15 dB is obtained for the operating frequency band (2.4–12.75 GHz) with the help of end-loaded meandered line connected to the ground plane.

The two-port antenna presented in [71] comprises a shared radiator with two shorting strips connected to the ground plane via holes at their end. High isolation is achieved for the impedance bandwidth from 3.3 GHz to 3.7 GHz through appropriate adjustment of the two feeding ports’ distance and by proper placement of shorting strips on the radiator. Compared to [71], the dual-feed MIMO antenna designed with one shared radiator and three shorting pins achieves a good isolation of less than −25 dB for the impedance bandwidth from 3.4 to 3.6 GHz [72]. Due to the insertion of shorting pins between two feeding ports and on the other sides as shown in Figure 11, the current distribution takes place everywhere on the antenna, which leads to good isolation.

In [73], a two-port PIFA with a common radiator placed at a height of 5 mm from the substrate having two feed plates placed perpendicular to each other as shown in Figure 12. A slot is etched on the ground plane which reduces the current flow between the two ports which results in improved isolation. The impedance bandwidth achieved is around 800 MHz from around 2.1 to 2.9 GHz. The ECC is below 0.005 and the MEG of the two ports is approximately equal to 1.

In [74], a common single circular patch radiator designed for UWB application with dimensions of 45 × 45 mm^2^ was proposed. An edged rectangular slot placed in the radiator increases the isolation between the two ports for the operating frequency range from 3 to 12 GHz. The measured ECC in terms of far field is less than 0.01. In [75], a circular-shaped common radiator is connected by two tapered coplanar waveguide feeding structures as shown in Figure 13. It consists of a ground plane printed on the bottom layer corner of the substrate and the circular radiator placed on the top layer. In this antenna design, the implemented common radiating element decreases the overall size of the antenna to dimensions of 41 mm × 41 mm and supports in creating the band notch elements in the radiator. Good isolation is achieved through the rectangular slot and inverted Y-shaped strip placed in the middle of the radiator and to the ground plane respectively. Wide bandwidth is obtained for UWB applications, which covers the frequency ranging from 2.9 GHz to 12 GHz.

As discussed in this section, a CPW-feed provides easy integration and placement of multiple antennas with shared radiator with existing MMIC technology on a single-sided PCB board. The objective is to have compact antennas, space being the primary concern in most of the applications. Thus, an alternative feeding technique is required, the ACS [65,66] feed antenna currently has been significantly explored and can still facilitate the basic antenna requirements. In [66], through proper placement of an I-shaped slot in the radiator along with a stub and a slot etched on ground plane achieve good isolation. In [67], by using a cross-shaped slot etched in a common radiator, mutual coupling is reduced, but the antenna performance is not identical for both ports due to asymmetrical radiator configuration seen from the two ports. In addition, most of the single element antennas reported in the literature are for UWB-MIMO applications and to achieve ample isolation between antenna ports, slots are used.

The usage of a slot on a shared radiator is one technique for obtaining effective isolation between the antenna ports. To control the effect of mutual coupling, a chain-shaped parasitic isolation structure is used for the common element MIMO antenna and ground plane [61]. Similarly, a circular patch is shared by two co-planar waveguide (CPW) feeds and mutual coupling is reduced with the help of a single slot etched in the patch [74] and by introducing a slot in patch and stub etched to the ground plane. In [75], two band-rejections are also achieved using elliptical split ring slots.

### 4.2. Single Element Four-Ports MIMO Antenna System

Two compact co-radiators with dual polarization, where each radiator shares two antenna elements, are discussed in [76]. A metal branch, expanded and etched in the two antenna elements at symmetric axis and which can be viewed as a reflector, decreases the current flow on the ground plane and reduces the electromagnetic coupling. The combination of a T-shaped slot on the patch with a pair of slits on the ground plane, and a stub attached to the ground plane results in good isolation. Further, the two slots placed in the ground plane stops the current in the ground from flowing to the other port. Dual polarization is also achieved by exciting the shared radiator with two perpendicular feeds. The four ports’ semi-elliptical monopole antennas with inverted L-shaped ground layer on the same plane are presented. The isolation is improved through polarization diversity and by inserting narrow slits to the degenerated ground plane, improves isolation at low frequencies. By implementation of inverted ‘L’ shaped ground layer, the ground plane acts as a reflector and operates as a directional antenna element as shown in Figure 14.

In [77], asymmetric coplanar strip (ACS)-fed multi-port antenna is designed with four antenna elements and two radiators in which the two antenna elements share a single radiator to provide overall dimensions of 36 × 36 mm^2^. Good isolation between the antenna elements is achieved and the operating bandwidth is also broadened by placing two I-shaped slots at the middle of the radiator and a rectangular patch attached on the bottom of the substrate, as shown in Figure 15. The surface current flowing between the ports is blocked by the stubs and reduces the mutual coupling. Since the ground plane presented here is asymmetric, the radiation pattern gets weaker at the higher frequency range.

In [78], a microstrip patch antenna having multi-cut design with four ports sharing a common radiator and the stepped ground is considered, as shown in Figure 16a. The energy released as surface current from the ports enters in the undesired directions. This causes mutual coupling and is reduced by etching multiple saw-tooth cuts at the corners of the shared rectangular patch. The overall size of the antenna is 75 × 75 × 1.6 mm^3^ and covers the frequency range from 4.96 to 5.5 GHz with gain varying between 2.4 and 5.5 dBi in the proposed band. A four-port shared radiator is further improved with better isolation and overall size of the antenna is 75 × 75 × 1.6 mm^3^ is presented in [79] as shown in Figure 16b. The designed stepped ground and the multi-cuts at the radiator patch controls the surface current direction which subsequently reduces the mutual coupling. The signal to noise ratio and back lobe radiation is also reduced through stepped rectangular design geometry. Good isolation is achieved for the entire frequency range from 4.4 GHz to 6.4 GHz. Low ECC of <0.04 is obtained and the gain of the designed antenna varied between 3.0 and 6.1 dBi.

In [80], four microstrip feedline monopole antenna connected to a shared radiator and ground plane is discussed. To achieve wideband, the rectangular-shaped common radiator is modified with stepped lines at the four corners as shown in the Figure 17a, to provide a better bandwidth of 1100 MHz. The four slots etched in each corner of the modified ground plane disturbs the current flow to achieve good isolation of better than 15 dB at the expected frequency range from 1.8 GHz to 2.9 GHz. The port coupling is also reduced via slot-loaded technique. Low measured ECC of < 0.1 and DG of about 10 dB are achieved.

In [81], the common element MIMO antenna with a notch-loaded circular radiator is fed by four identical modified feedlines, suitable for Wi-Fi applications, is proposed as shown in Figure 17b. The uniformity of the circular shaped radiator reduces the surface current, and the fringing field is formed by choosing the modified circular geometry concept. The surface current direction is further disturbed by the octagonal notches loaded in the circular radiator and thus the mutual coupling is reduced. The parasitic element placed diagonally between the ground plane enhances the inter-port isolation and ECC less than 0.5 is achieved.

The differentially-fed MIMO antenna with dual-band is discussed in [82]. It consists of four ports sharing a common radiator and a square ring ground, suitable for WiMAX application, as shown in Figure 18a. Four differential ports arranged perpendicular to each other connected to virtual ground provides dual polarization and good isolation is achieved for dual frequency bands from 3.2–3.87 GHz and 5.35–5.83 GHz. Since the designed antenna is CPW differential fed, both the radiating patch and the virtual ground is printed on the same layer of the substrate. This makes the fabrication process easy and suitable for integration with RF front-end circuits. In [83], a monopole antenna with four ports sharing a common radiator and a ground plane with slots on all four edges was proposed covering the frequency ranging from 2 GHz to 3 GHz. The steps-shape common radiator patch placed on the top layer and frame-shape ground plane minimizes the mutual coupling effect among the ports which provides better isolation of less than −12 dB as shown in Figure 18b. With the partial removal of ground plane on the bottom plate, the bandwidth is enhanced and ECC below 0.5 is achieved. The common radiator with four port MIMO antenna, discussed in [84], consists of a common radiating element and a square ring ground plane. Good isolation is achieved through space diversity and by etching the stubs in the ground plane, to achieve isolation of less than −15 dB and measured return loss of −10 dB is achieved for frequency range from 4.5 GHz to 7 GHz.

## 5. Summary

In this paper, a concise review on different multiport single antenna element systems has been presented. Comparison has been done based on antenna size, substrate, number of ports, ECC, antenna gain, and radiation efficiency, and summarized in Table 2 and Table 3 for the compared MIMO antenna designs.

Designing a compact common radiator MIMO antenna system with strong isolation between antenna ports is important for future wireless communication. Several isolation techniques considered for MIMO antenna design are discussed in this review paper. By implementing DGS and CSRR technique, the MIMO antenna provides wide band with lower ECC values. The simple structure with a significantly improved isolation is achieved through a neutralization line technique and parasitic elements provide size reduction with improved efficiency. Similarly, other mutual coupling reduction techniques discussed in the literature offer better isolation, but maintaining improvement in efficiency and size reduction could be a challenge for shared radiator MIMO antenna design.

The implementation of a slot on a common radiating patch is another technique for obtaining effective isolation between antenna ports but the antenna performance is affected. Similarly, CPW-fed structure provides easy integration but increase in space between the antenna elements leads to less suitability for portable devices. Compared to this CPW-fed structure, ACS structure provides good performance in decreasing the overall size, due to its half-ground identity of antennas. Additionally, the ACS-fed structure provides better performance with additional features such as single lateral ground-plane, wide bandwidth, and easy to integrate with portable devices.

But these antenna configurations suffer minor variations in radiation characteristics behavior due to the asymmetrical nature of the antenna. Therefore, these antennas would be implemented for those applications where the radiation pattern does not substantially affect the performance of the antenna. Besides this, most of the single common element antennas reviewed here are suitable for UWB-MIMO applications.

The shared radiator in a MIMO system is an effective method to retain the compactness of the antenna design for the portable devices as compared to separate antenna elements. This also provides unique radiation pattern characteristics, but to design with good isolation is difficult. Hence this paper presents several antennas designs and mutual coupling reduction techniques which would be helpful in designing antenna for future communication systems and in achieving the required performance parameters of the MIMO antennas. In all the above cases, most of the single element antennas reported in the literature are for UWB-MIMO and Wi-Fi/WLAN applications. Hence, it is important in the future, to consider efficient technique to design MIMO antenna with shared radiator for 5G and portable devices.

## Figures and Tables

**Figure 1 sensors-23-00747-f001:**
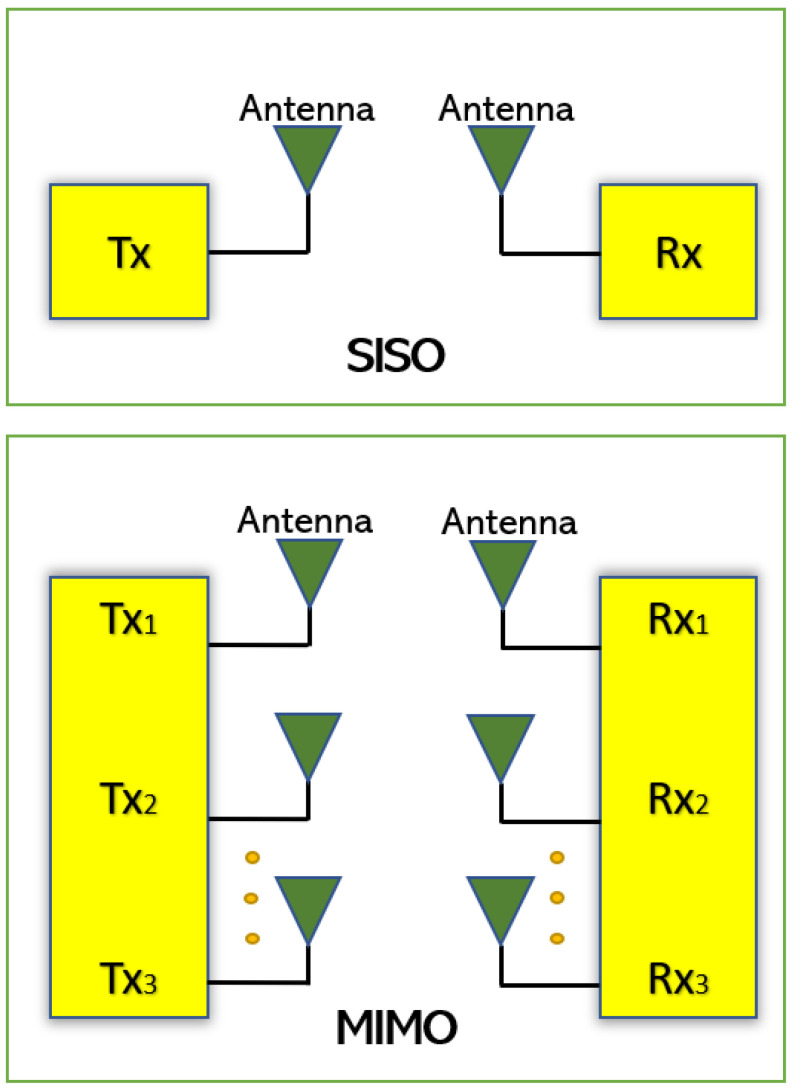
SISO and MIMO Antenna system.

**Figure 2 sensors-23-00747-f002:**
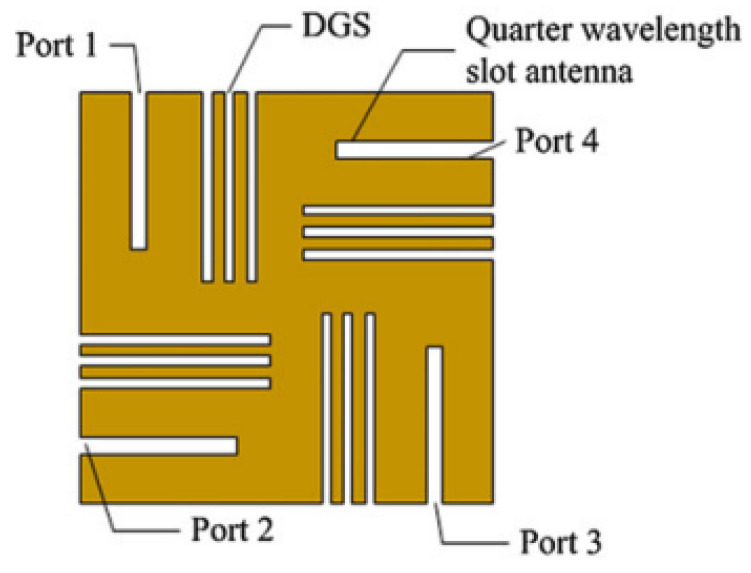
Orthogonally polarized slot antenna DGS [16].

**Figure 3 sensors-23-00747-f003:**
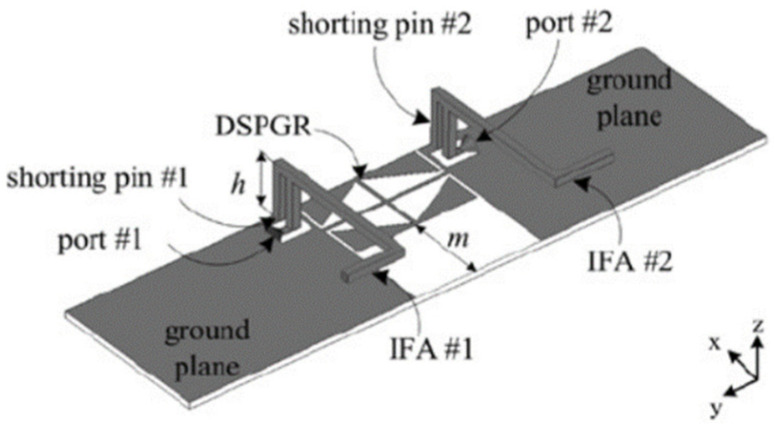
Diamond-shaped patterned ground resonator [23].

**Figure 4 sensors-23-00747-f004:**
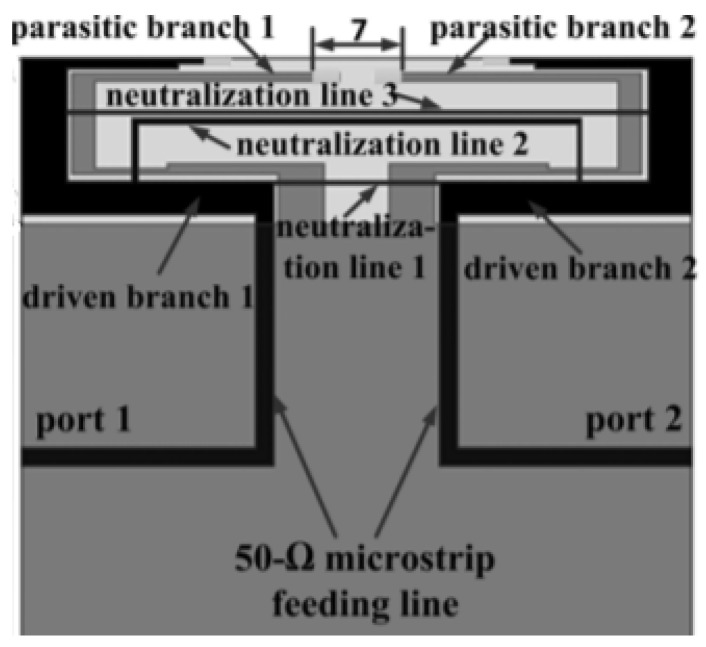
Neutralization line technique [29].

**Figure 5 sensors-23-00747-f005:**
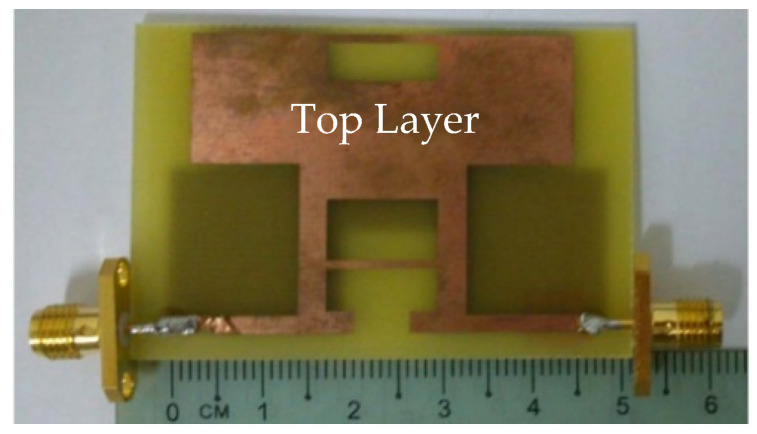
Fabricated prototype with T-shaped slot [60].

**Figure 6 sensors-23-00747-f006:**
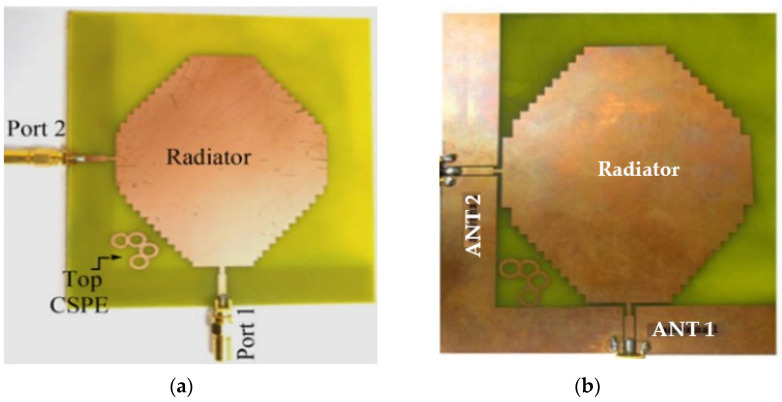
(**a**) Fabricated prototype with CPSE [61] (**b**) CPW-fed fabricated prototype with CPSE [62].

**Figure 7 sensors-23-00747-f007:**
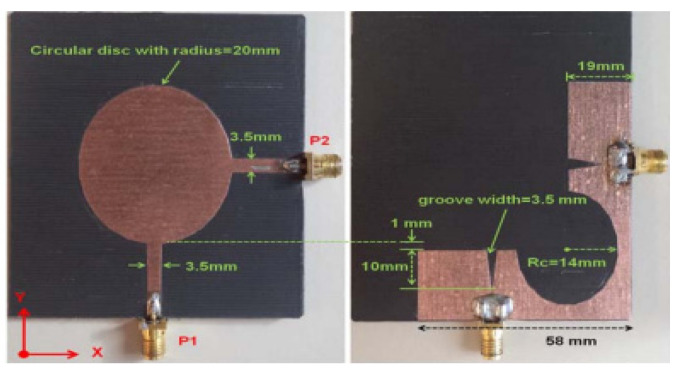
Shared radiator with circular cut at ground [63].

**Figure 8 sensors-23-00747-f008:**
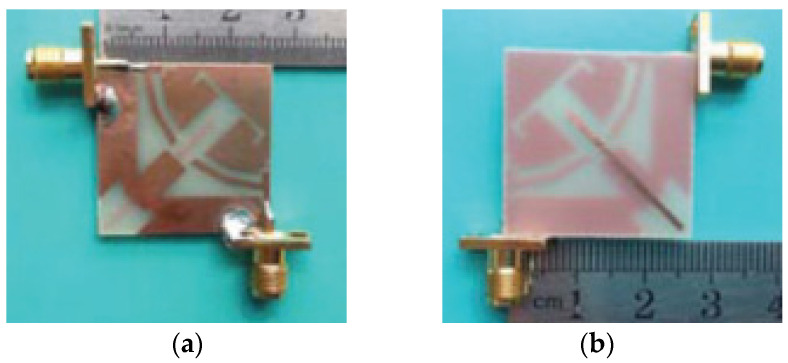
(**a**) Top layer of ACS-fed structure with common radiator (**b**) Bottom layer [66].

**Figure 9 sensors-23-00747-f009:**
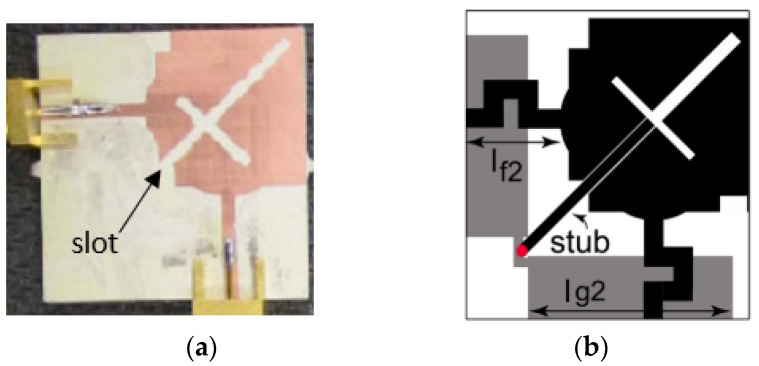
(**a**) Slots etched in the common radiator antenna design [67], (**b**) Open shunt stub placement and etched T-shaped slots in the common radiator antenna design [68].

**Figure 10 sensors-23-00747-f010:**
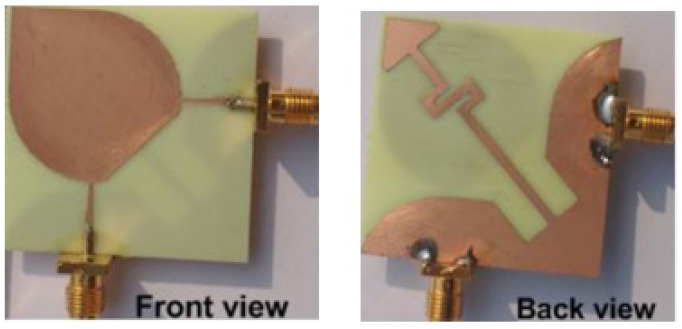
Front and bottom view of two port MIMO antenna design with common radiator [70].

**Figure 11 sensors-23-00747-f011:**
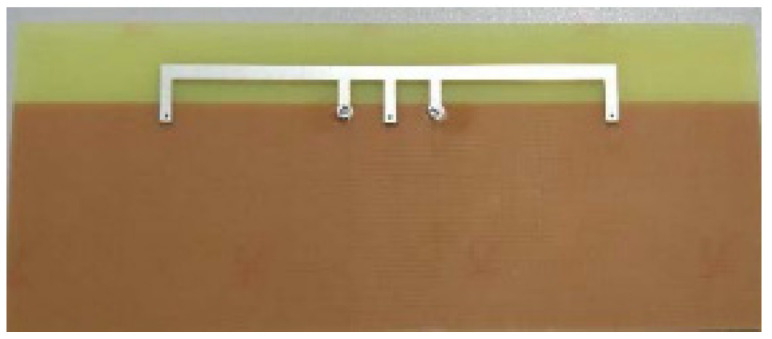
Fabricated two port shared radiator with shorting pins [72].

**Figure 12 sensors-23-00747-f012:**
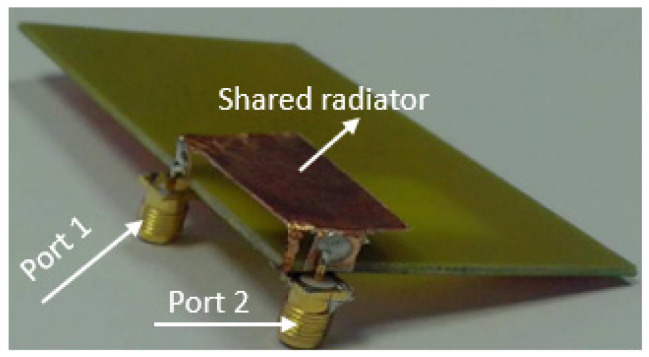
Fabricated two port PIFA MIMO antenna design with shared radiator [73].

**Figure 13 sensors-23-00747-f013:**
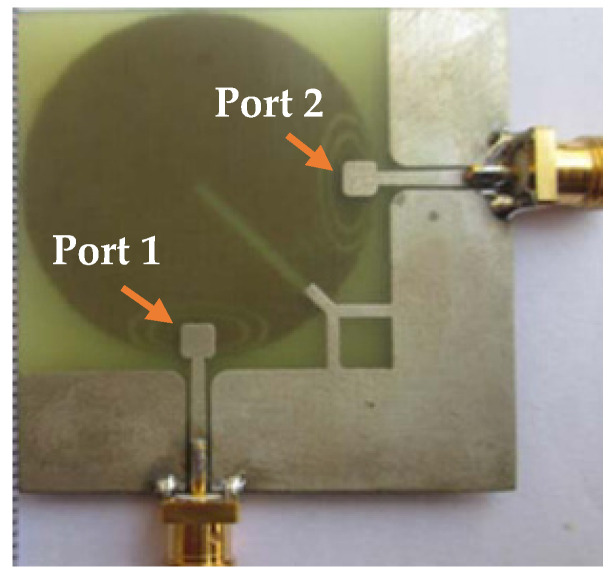
Circular patch shared radiator [75].

**Figure 14 sensors-23-00747-f014:**
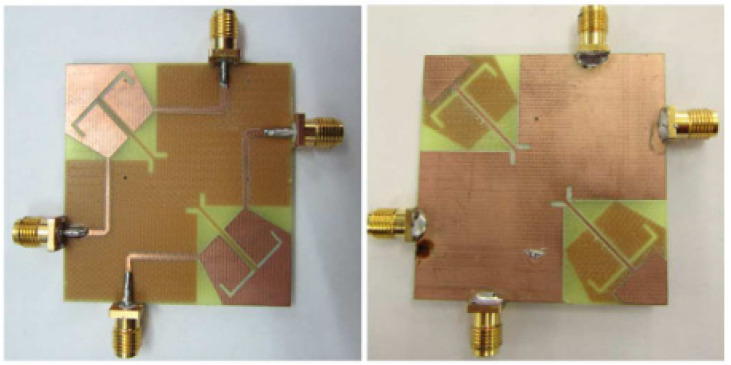
Compact co-radiator with dual polarization [76].

**Figure 15 sensors-23-00747-f015:**
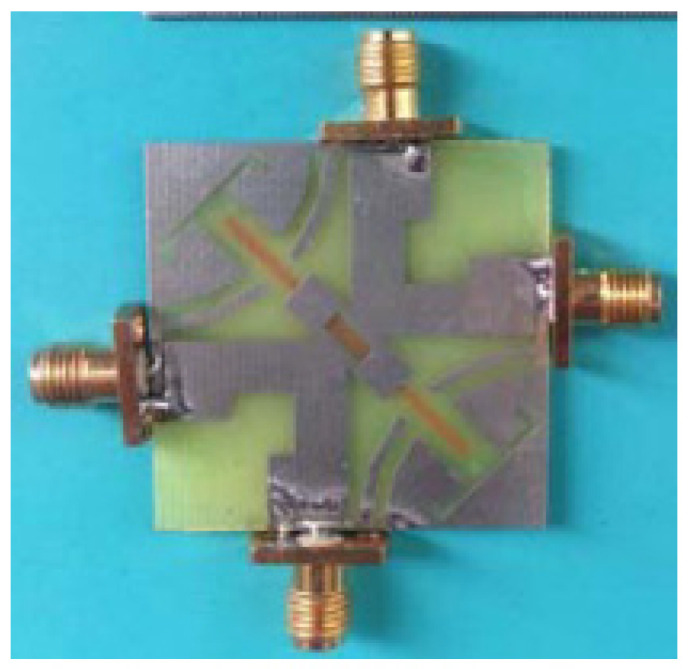
ACS-fed 4 port antenna [77].

**Figure 16 sensors-23-00747-f016:**
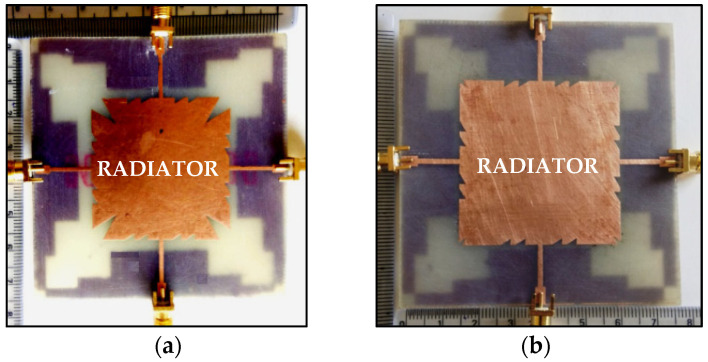
(**a**) Four ports multi-cut shared radiator and stepped ground [78], (**b**) Four ports multi-cut shared radiator and stepped ground [79].

**Figure 17 sensors-23-00747-f017:**
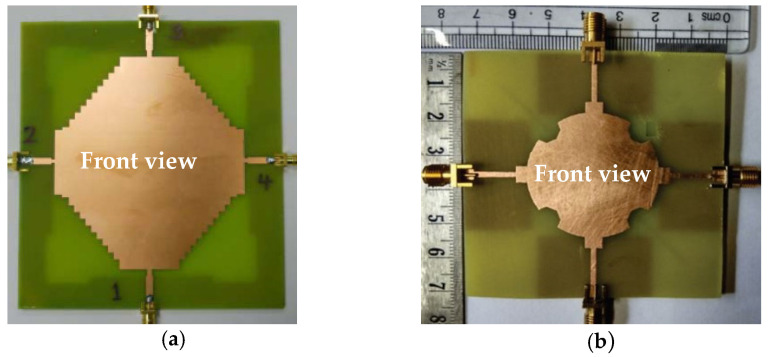
(**a**) Stepped line common radiator prototype [80], (**b**) Notch-loaded circular radiator [81].

**Figure 18 sensors-23-00747-f018:**
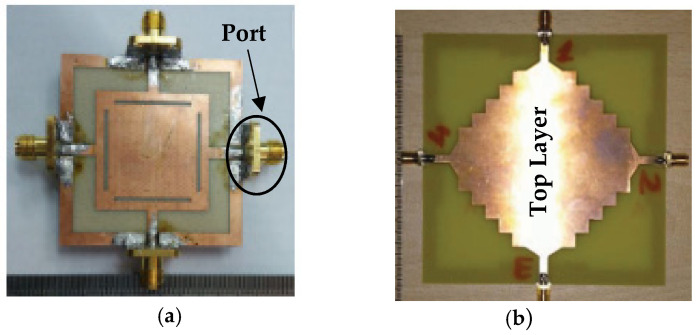
(**a**) Common radiator with square ring ground [82], (**b**) Quad-port MIMO antenna [83].

**Table 1 sensors-23-00747-t001:** Acronyms.

Acronyms	Meanings
SISO	Single Input Single Output
MIMO	Multiple Input Multiple Output
TARC	Total Active Reflection Coefficient
ECC	Envelop Correlation Coefficient
DG	Diversity Gain
DGS	Defected Ground Structure
DMN	Decoupling and matching network
CDRN	Coupled Resonator Decoupling Network
CSRR	Complementary Split Ring Resonators
ACS	Asymmetric Coplanar Strip
CPW	Co Planar Waveguide
CE	Common Element
PIFA	Planar Inverted-F antenna
iMAT	Isolated Mode Antenna Technology
EBG	Electromagnetic Bandgap
MEG	Mean Effective Gain
WiFi	Wireless Fidelity
UWB	Ultra-Wide Band
LTE	Long-Term Evolution
WiMax	Worldwide Interoperability for Microwave Access

**Table 2 sensors-23-00747-t002:** Shared radiator antenna with two ports.

References	Frequency (GHz)	Size/Material (mm^2^)	Isolation Technique	Isolation (dB)	ECC	Gain (dBi)	No. of Ports/Application
[60]	2.3–2.6	48 × 33.8/FR4	Slots/Ground slot	−16	0.1	1.2	2/WLAN (2.4 GHz)
[61]	2.3–2.9	105 × 125/FR4	CSPE rings	−15	0.15	5	2/WiFi (2.4 GHz) and LTE (2.6 GHz)
[62]	1.2–1.55; 2.3–2.7	110 × 130/FR4	CSPE rings	−15	0.15	3.3–5.6	2/WiFi (2.4 GHz), WiMAX (2.3 and 2.5 GHz), and LTE (1.5, 2.6 GHz)
[63]	2–6	64 × 64 mm/RT Duroid 5880	Ground plane circular slot	−15	0.02	3.2–6.1	2/MIMO applications
[66]	3.1–10.6	26 × 26/FR4	I and U-shaped stub	−15	0.02	2.2, 3.5, 2	2/Ultra-Wide Band
[65]	2.6–11	28.5 × 28.5/FR4	Rectangular stub	−15	0.01	1.5–3.7	2/Ultra-Wide Band
[67]	3–11	27 × 27/Rogers TMM4	Slot	−15	0.01	3	2/Ultra-Wide Band
[68]	3.1–10.6	22 × 24.3/Rogers TMM4	Slots, open shut stub	−15	0.02	2–5.5	2/Ultra-Wide Band
[70]	2.4–12.75	39 × 39/FR4	Stub, meandered line	−15	0.02	4.92	2/Ultra-Wide Band
[71]	3.35–3.65	90 × 50/FR4	Positioning shorting strips	−20	0.1	3.5	2/MIMO applications
[72]	3.4–3.7	130 × 50/FR4	3 shorting strips	−25	0.017	4	2/MIMO applications
[73]	2.1–2.9	45 × 100/FR4	Slot at ground plane	−14	0.005	6.4	2/2.45-GHz WLAN band and WiMAX band (2.5–2.7 GHz)
[74]	3–12	45 × 45	Slot	−17	0.01	−4 to 2	2/Ultra-Wide Band
[75]	3–12	41 × 41	Slot	−14	0.01	2	2/ Ultra-Wide Band

**Table 3 sensors-23-00747-t003:** Shared radiator antenna with four ports.

References	Frequency (GHz)	Size (mm^2^)	Isolation Technique	Isolation (dB)	ECC	Gain (dBi)	Number of Ports/Application
[76]	3–11	48 × 48	T shaped slot	−17	0.02	2–5	4/Ultra-Wide Band
[77]	3.1–10.6	36 × 36	Stub at ground plane	−15	0.02	1.5–4	4/Ultra-Wide Band
[78]	4.96–5.5	75 × 75	Defected patch and ground	−12	0.03	2.4–5.5	4/WLAN (5.2)
[79]	4.4–6.4	75 × 75	Defected patch and ground	−13	0.04	3–6.1	4/Sub 6 GHz applications
[80]	1.8–2.9	120 × 140	Slots at ground, DGS	−15	0.07	6	4/LTE (Wireless Access Point), WiFi (2.4)
[81]	2.34–2.56	72 × 72	Diagonal parasitic element	−12	0.01	2	4/WiFi (2.4)
[82]	3.5, 5.5	45 × 45	Slot technique	−35	0.02	2–5	4/WiMAX (3.5,5.5)
[83]	2–3	108 × 108	Slot technique	−12	0.02	5.9–6.2	4/LTE (2.1/2.3/2.6 GHz), Wi-Fi (2.4)
[84]	4.5–7	95 × 95	Ground stub	−20	0.01	5	4/WLAN (5.2,5.8), WiMAX (5.5)

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
