# Peer review of "Multiport Single Element Mimo Antenna Systems: A Review"

_sensors, 2023, doi:10.3390/s23020747_

Round 1

Reviewer 1 Report

In this paper, 6the authors investigate an in-depth review of different MIMO antenna designs with common radiators covering various antenna design aspects such as isolation techniques, gain, efficiency, etc. Meantime, the mathematical modeling of MIMO and different isolation techniques are discussed, and a comparative analysis of different shared radiator antenna designs is also precisely covered.

This presentation of the work is good.

However, I have a few following concerns which are needed to be addressed.

Minor Comments:

1.    A detailed discussion is required on the MIMO antenna, which is the main theme of the paper, in the Introduction Section to further justify and strengthen the study. A few papers are listed below:

a.    https://doi.org/10.3390/s22155531

b.    https://doi.org/10.3390/mi11121083

c.     https://doi.org/10.3390/mi11110956

d.    https://doi.org/10.3390/electronics9061031

2.    Use one format throughout the paper for all abbreviations

3.    Improve the quality of figures such as Figs. 17 and 18. 

Author Response

Dear Reviewer,

The authors would like to thank you for your efforts in reviewing the manuscript and for your valuable and constructive comments and fruitful observations that helped in improving the quality of the manuscript to a publishable standard. Detailed below are the responses to the reviewer’s comments and suggestions. Reviewer’s questions and comments are in BLACK and the authors’ answers and comments are in BLUE. The modifications and additions are made appear in RED in the manuscript.

In this paper, the authors investigate an in-depth review of different MIMO antenna designs with common radiators covering various antenna design aspects such as isolation techniques, gain, efficiency, etc. Meantime, the mathematical modeling of MIMO and different isolation techniques are discussed, and a comparative analysis of different shared radiator antenna designs is also precisely covered.

This presentation of the work is good.

However, I have a few following concerns which are needed to be addressed.

We would like to thank the Reviewer for the careful review and the valuable comments that has helped in improving the quality of the manuscript submitted. We have carefully evaluated your comments, now enumerated below together with our responses.

Point 1: A detailed discussion is required on the MIMO antenna, which is the main theme of the paper, in the Introduction Section to further justify and strengthen the study.

A few papers are listed below:

  1. https://doi.org/10.3390/s22155531
  2. https://doi.org/10.3390/mi11121083
  3. https://doi.org/10.3390/mi11110956
  4. https://doi.org/10.3390/electronics9061031

Response 1: The authors would like to thank the reviewer for this valuable comment. We have included the mentioned references with the related details in the manuscript which are highlighted in Red in the manuscript.

Point 2: Use one format throughout the paper for all abbreviations

Response 2: The authors strongly agree with the reviewer, and this has been considered. We have thoroughly gone through the paper again and all such issues are not resolved which are highlighted in Red in the manuscript.

Point 3: Improve the quality of figures such as Figs. 17 and 18. 

Response 3: The authors highly appreciate this valuable comment and careful observation of the Reviewer. The pictures in Figs. 17 and 18 are now updated in the manuscript. Thanks!

Reviewer 2 Report

The authors provided a review of MIMO antenna designs, including the mathematical concepts of MIMO, their isolation techniques and performance. This review is written in a logical and clear way, and has informative figures and detailed references. I recommend accept in present form.

Author Response

The authors provided a review of MIMO antenna designs, including the mathematical concepts of MIMO, their isolation techniques and performance. This review is written in a logical and clear way and has informative figures and detailed references. I recommend accept in present form.

We would like to thank the reviewer for the careful review and the valuable comments that has helped in improving the quality of the manuscript submitted and acknowledging the efforts we have put to make this manuscript a good, addition to the literature.

Reviewer 3 Report

The article is very interesting and valuable. It deals with many issues in the field of multi-element MIMO antennas. But this is only review. This paper has no presentad any authors researches. But the review is needed, cause very big number works and methods for designing antennas.

MIMO antennas are used in radiocommunication systems as mobile phones, Wi-Fi or UWB systems. If paper is written for Special Issue about IoT and RFID, should be noted that is no any information about MIMO in these fields.

The paper is well organized and well written. However, the authors did not avoid many minor mistakes.

1. Please standarize matrix notation. Sometimes matrix is shown in bold (as matrix H), sometimes in parentheses (as matrix S).

2. Equation 1 is bold letters. Why? Maybe better will be change to standard letter. What is mean dot between H matrices in this equation?

3. Equation 3: Gamma parameter is the same as in equation 2? What is mean indexes in gamma in equation 3?

4. Please review the work for the abbreviations used. It is good to follow the rule that the first time an abbreviation is used, it is translated. This is not always the case at work (for example ECC in abstratct, MEG on page 3). I suggest to sfhit the Table 1 earlier. In the text should be used reference to this table.

5. Word "Figure" sometimes is bold letter sometimes is normal.

6. I suggest to create subchapter from text chapter 4, before 4.1. I suggest put at the end of 4.1 subchapter some words as reference to Table  2.

7. Figure 8 is bad quality. Change for better picture is needed.

8. Table 2. Pleease, give unit to all columns. Is gain value is dB? This column has no unit. Please, give the unit for the size. For first line exist mm2 - this is square.  Some words about LTE band is needed inside table. LTE can work in many frequency bands, from 400 MHz up to 6 GHz. The same situation is Wi-Fi. Wi-Fi in 2.4 GHz or 5.8 GHz or in both bands? In table, ref. [64], is only Rogers material. Rogers is the family of the substrate. Please, give full name of the substrate. 

Author Response

Dear Reviewer,

The authors would like to thank you for your efforts in reviewing the manuscript and for your valuable and constructive comments and fruitful observations that helped in improving the quality of the manuscript to a publishable standard. Detailed below are the responses to the reviewers’ comments and suggestions. Reviewers’ questions and comments are in BLACK and the authors’ answers and comments are in BLUE. The modifications and additions made appear in RED in the manuscript.

The article is very interesting and valuable. It deals with many issues in the field of multi-element MIMO antennas. But this is only review. This paper has no presented any authors researches. But the review is needed, cause very big number works and methods for designing antennas.

MIMO antennas are used in radiocommunication systems as mobile phones, Wi-Fi or UWB systems. If paper is written for Special Issue about IoT and RFID, should be noted that is no any information about MIMO in these fields.

The paper is well organized and well written. However, the authors did not avoid many minor mistakes.

We would like to thank the Reviewer for the careful review and the valuable comments that helped in improving the quality of the manuscript submitted. We have carefully evaluated the comments, now enumerated below together with our responses.

Point 1 Please standardize matrix notation. Sometimes matrix is shown in bold (as matrix H), sometimes in parentheses (as matrix S).

Response 1: The authors would like to thank the reviewer for this valuable comment. We have thoroughly gone through the paper again and all such issues are now resolved. The notations are now standardized and updated in the manuscript. Thanks!

Point 2 Equation 1 is bold letters. Why? Maybe better will be change to standard letter. What is mean dot between H matrices in this equation?

Response 2: Authors highly appreciate this valuable comment and careful observation of the Reviewer. All the equations are changed to standard letters and updated in the manuscript.

Thanks!

Point 3 Equation 3: Gamma parameter is the same as in equation 2? What is mean indexes in gamma in equation 3?

Response 4: The authors would like to thank the reviewer for this valuable comment. The equation 3 is for a given excitation [a]. Whenever, the TARC value is equal to zero, it indicates that all the delivered power is radiated and when it is equal to one, all the power is either reflected back or goes to the other ports.

Point 4 Please review the work for the abbreviations used. It is good to follow the rule that the first time an abbreviation is used, it is translated. This is not always the case at work (for example ECC in abstract, MEG on page 3). I suggest shifting the Table 1 earlier. In the text should be used reference to this table.

Response 4: The authors strongly agree with the reviewer, and this has been considered. We have thoroughly gone through the paper again and all such issues are now resolved. The related changes are modified accordingly in the manuscript.

Point 5 Word "Figure" sometimes is bold letter sometimes is normal.

Response 5: Authors highly appreciate this valuable comment and careful observation of the Reviewer. All the words with “figure” are changed to “Fig.” in the manuscript highlighted in Red.

Thanks!

Point 6 I suggest creating subchapter from text chapter 4, before 4.1. I suggest put at the end of 4.1 subchapter some words as reference to Table 2.

Response 6: The authors would like to thank the reviewer for this valuable comment. The content discussed in the chapter 4 before subchapter 4.1 are about existing MIMO antennas with multielement. Because of this, it was not considered as a subchapter. Thanks!

Point 7 Figure 8 is bad quality. Change for better picture is needed.

Response 7: Authors highly appreciate this valuable comment and careful observation of the Reviewer. The picture in Fig. 8 is updated with better quality and updated in the manuscript.

Thanks!

Point 8 Table 2. Please, give unit to all columns. Is gain value is dB? This column has no unit. Please, give the unit for the size. For first line exist mm2 - this is square.  Some words about LTE band is needed inside table. LTE can work in many frequency bands, from 400 MHz up to 6 GHz. The same situation is Wi-Fi. Wi-Fi in 2.4 GHz or 5.8 GHz or in both bands? In table, ref. [64], is only Rogers material. Rogers is the family of the substrate. Please, give full name of the substrate. 

Response 8: Authors highly appreciate this valuable comment and careful observation of the Reviewer. The frequency bands are discussed in column 2 of Table 1 and Table 2. All the columns are now updated with the units and the related changes are modified accordingly in the manuscript. Thanks!

Round 2

Reviewer 1 Report

Well done! My previous comments are properly addressed. I have no more comments.

The paper is recommended for publication.